# Pancreatic Ductal Adenocarcinoma: The Characteristics of Contrast-Enhanced Ultrasound Are Correlated with the Hypoxic Microenvironment

**DOI:** 10.3390/diagnostics13203270

**Published:** 2023-10-20

**Authors:** Lan Wang, Ming Li, Tiantian Dong, Yuanyuan Li, Ci Yin, Fang Nie

**Affiliations:** 1Ultrasound Medical Center, Lanzhou University Second Hospital, Cuiyingmen No. 82, Chengguan District, Lanzhou 730030, China; wanglan19@lzu.edu.cn (L.W.); limingshy@163.com (M.L.); dongtt21@lzu.edu.cn (T.D.); liyy21@lzu.edu.cn (Y.L.); yin-ci@outlook.com (C.Y.); 2Gansu Province Clinical Research Center for Ultrasonography, Lanzhou 730030, China

**Keywords:** ultrasonography, contrast agent, pancreatic neoplasms, hypoxia, tumor microenvironment

## Abstract

A hypoxic microenvironment is associated with an increased risk of metastasis, treatment resistance and poor prognosis of pancreatic ductal adenocarcinoma (PDAC). This study aimed to identify contrast-enhanced ultrasound (CEUS) characteristics that could predict the hypoxic microenvironment of PDAC. A total of 102 patients with surgically resected PDAC who underwent CEUS were included. CEUS qualitative and quantitative characteristics were analyzed. The expression of hypoxia-inducible factor-1α (HIF-1) and glucose transporter-1 (GLUT1) were demonstrated by immunohistochemistry. The associations between CEUS characteristics and the HIF-1α and GLUT1 expression of PDACs were evaluated. We found that HIF-1α-high PDACs and GLUT1-high PDACs had a larger tumor size and were more prone to lymph node metastasis. There was a significant positive linear correlation between the expression of HIF-1α and GLUT1. CEUS qualitative characteristics including completeness of enhancement and peak enhancement degree (PED) were related to the expression of HIF-1α and GLUT1. A logistic regression analysis showed that tumor size, lymph node metastasis, incomplete enhancement and iso-enhancement of PED were independent predictors for HIF-1α-high PDACs and GLUT1-high PDACs. As for quantitative characteristics, HIF-1α-high PDACs and GLUT1-high PDACs showed higher peak enhancement (PE) and wash-in rate (WIR). CEUS can effectively reflect the hypoxia microenvironment of PDAC, which may become a noninvasive imaging biomarker for prognosis prediction and individualized treatment.

## 1. Introduction

Pancreatic ductal adenocarcinoma (PDAC) is typically characterized by multitudinous and severe hypoxic regions [1]. The hypoxic microenvironment, which is considered to be one of the independent prognostic factors for PDAC, regulates many important biological hallmarks of cancer, ranging from tumor cell differentiation, tumor progression, and metabolic reprogramming to chemotherapy resistance [2,3]. The specialized hypoxic microenvironment is commonly accepted to be the main reason for the poor efficacy of many treatments for PDAC [2]. Hypoxia-targeted therapies provide new opportunities to overcome the resistance to chemotherapy and radiotherapy and enhance therapeutic efficacy, and a reliable imaging biomarker for hypoxia could help select appropriate patients and evaluate the therapeutic efficacy. Contrast-enhanced ultrasound (CEUS) is performed with a blood pool contrast agent that demonstrates real-time dynamic perfusion of tissue and has proven to be an accurate imaging method for identifying pancreatic tumor vascularization that is closely related to the tumor hypoxic microenvironment [4,5]. A compromised oxygen supply due to poor vascularization is one of the main causes of the hypoxic microenvironment of PDAC, and tumor hypoxia will further induce transcriptional activation of genes to promote neoangiogenesis [3,6].

Cells can respond to reduced oxygen levels through the hypoxia-inducible transcription factor 1 (HIF-1) [7]. HIF-1 is composed of hypoxic response factor HIF-1α and HIF-1β. HIF-1α is a key transcription factor that is induced by hypoxia and regulated by a proline hydroxylase. Previous studies [7,8] have reported that HIF-1α was overexpressed in various solid tumors and is associated with cell proliferation and unfavorable patient prognosis. HIF-1α plays pivotal roles in the hypoxia-induced therapy resistance of PDAC, and many hypoxia-targeted therapies work by inhibiting the expression of HIF-1α or HIF-1α-associated molecules such as glucose transporter-1 (GLUT1). GLUT1, which increases under hypoxia, can mediate cellular glucose uptake and, thus, facilitates anaerobic glycolysis, making it an important factor for the proliferation of cancer cells under hypoxic conditions [9]. GLUT1 expression has been reported to be significantly associated with prognosis in pancreatic cancer [10,11]. Therefore, we chose to assess the expression of HIF-1α and GLUT1 to measure the hypoxic microenvironment of PDAC, and to identify CEUS characteristics that could noninvasively evaluate the expression level of HIF-1α and GLUT1. Our study may be the first to report CEUS characteristics as indicators of hypoxia in PDAC, which can be easily incorporated into clinical practice for prognosis prediction and individualized treatment.

## 2. Materials and Methods

### 2.1. Study Subjects

This retrospective study was approved by our institutional review board (No: 2023A-283). Informed consent was waived because of the retrospective nature of the study. A total of 102 patients who received pancreatic CEUS exams before surgery and were pathologically diagnosed with PDAC using samples obtained via operation in our hospital from May 2019 to August 2023 were enrolled in this study. The inclusion criteria were as follows: (1) age ≥ 18; (2) confirmed to have PDAC via surgery samples; (3) underwent preoperative CEUS imaging of the pancreas within 1 month before surgery. The exclusion criteria were as follows: (1) preoperative CEUS absent; (2) without surgical pathology; (3) unqualified CEUS images; (4) incomplete or missing clinical information. Among the 102 cases, 4 (3.9%) patients underwent surgery after neoadjuvant chemotherapy; 88 (86.3%) patients had resectable PDAC, and 14 (13.7%) patients had borderline resectable PDAC; 14 (13.7%) cases had vascular involvement, including solely splenic vein involvement in 6 cases, solely superior mesenteric vein involvement in 1 case, solely superior mesenteric artery involvement in 1 case, combined portal vein and splenic vein involvement in 2 cases, combined superior mesenteric vein and superior mesenteric artery involvement in 2 cases, and combined splenic vein and splenic artery involvement in 2 cases; 84 (82.4%) cases underwent pancreaticoduodenectomy, 15 (14.7%) underwent distal pancreatectomy, and 3 (2.9%) underwent total pancreatectomy; 92 (90.2%) patients underwent R0 resection, and 10 (9.8%) patients underwent R1 resection; postoperative pancreatic fistula (POPF) occurred in 19 (18.6%) patients, postoperative bile leakage occurred in 7 (6.9%) patients, postpancreatectomy hemorrhage (PPH) occurred in 8 (7.8%) patients, and delayed gastric emptying (DGE) occurred in 7 (6.9%) patients. The mean duration of hospital stay was 27 days (range, 9–101), and the perioperative mortality within 30 and 60 days after pancreatectomy was 2% and 5%, respectively. 

### 2.2. CEUS Study Protocol

Patients were required to fast for at least eight hours prior to the scan, and all CEUS examinations were performed on the basis that the lesions can be clearly displayed on the corresponding gray-scale ultrasound (US). All US scans were carried out with ACUSON Sequoia (Siemens Medical Solutions, Malvern, PA, USA) equipped with convex array probe 5C1 (1–5 MHz) or Philips IU22/EPIQ 7 (Philips Healthcare, Hong Kong) equipped with convex array probe C5-1 (1–5 MHz). Low mechanical index (<0.08) imaging mode was used. Sulfur-hexaflfluoride microbubbles (SonoVue, Bracco, Milan, Italy) were used as the contrast agent, which was dissolved in 5 mL saline to make a microbubble suspension. Then, a fast bolus injection of 1–2 mL suspension followed by a 5 mL saline flush was injected into the antecubital vein. Each lesion was continuously observed for 120 s, and its enhancement characteristics were compared with the surrounding pancreatic parenchyma. The CEUS scan was divided into the arterial phase (<30 s) and the venous phase (31–120 s) after the microbubble suspension was injected. The imaging of all the phases was stored in the device for further analysis.

### 2.3. CEUS Analysis

#### 2.3.1. Qualitative Analysis

Two radiologists independently and blindly evaluated the CEUS images, and any divergent opinions were discussed until a consensus was reached. The following CEUS characteristics were evaluated: in the arterial phase, enhancement that appeared in the lesion earlier than, equal to or later than the pancreatic parenchyma was defined as early, synchronous and late wash-in, respectively; centripetal enhancement was defined as the contrast agents developing from the periphery to the center of the lesion; according to our clinical experiences and relevant study [12], penetrating vessels were defined as thin vessels branched around/in or penetrating the tumor; we divided lesions into complete enhancement and incomplete enhancement based on the completeness of enhancement, and we further divided incomplete enhancement into enhancement region ≤ 50% and enhancement region > 50%. We classified the peak enhancement degree (PED) as iso- or hypoenhancement in comparison to the normal pancreatic tissue (the following characteristics were also considered to be isoenhancement: 1. When the lesion showed heterogeneous enhancement with both iso- and hypo-enhancement, the isoenhancement area was >50%; 2. For solid-cystic lesions, the solid component manifested isoenhancement). In the venous phase, the washout of the lesion earlier than, equal to or later than that of the pancreatic parenchyma was defined as early, synchronous and late washout, respectively.

#### 2.3.2. Quantitative Analysis

Quantitative perfusion analysis software (VueBox, Bracco) version 4.3 was used to investigate CEUS video clips stored in the device. Regions of interest (ROI) were placed manually in the strongest enhancement of the suspected tumor. The following ROIs parameters from the quantification tool box were collected [13]: peak enhancement (PE, [a.u]), the maximum intensity of contrast medium signal; Wash-in Area Under the Curve (WiAUC, [a.u]), the sum of all amplitudes inside the range from the beginning of the curve rising to the highest point of the curve; rise time (RT, [s]), the time interval between first arrival of contrast medium and time of contrast medium peak intensity; time to peak (TTP, [s]), the time from the start of the injection to the maximum intensity of the curve; wash-in rate (WIR, [a.u]), the slope of the wash-in (ascending). 

#### 2.3.3. Clinicopathological Analysis 

Immunohistochemical Staining: Tissue sections (4 μm) were deparaffinized in xylene and rehydrated for 5 min per session in a graded series of 100, 95, 85 and 75% alcohol. Antigen retrieval was performed by heating in 0.01 M citrate buffer for 16 min in a microwave oven and cooled for 30 min. Endogenous peroxidase activity was blocked with 3% hydrogen peroxide (H_2_O_2_) in methanol and non-specific binding sites were blocked with protein-blocking solution (5% normal horse and 1% normal goat serum). Primary antibodies against HIF-1α (1:100 dilution, Proteintech, Rosemont, IL, USA), GLUT1 (1:200 dilution, Immunoway, Plano, TX, USA) were added and sections were incubated over night at 4 °C. Then, the sections were treated with secondary antibody and incubated with streptavidin-peroxidase (SP) complex (Maixin Biotechnology, Fuzhou, China) for 40 min at room temperature. Binding sites were visualized with 3,3-diainobenzidine (DAB) (Maixin Biotechnology) after 1 min incubation. The brown areas were the positive areas for HIF-1α and GLUT1 staining. For quantitative analysis of immunostaining intensity, integrated optical density (IOD) was employed using an image analyzer (Image Pro Plus 6.0). Digitally fixed images were analyzed at ×200 magnification using a light microscope (Olympus, Tokyo, Japan) equipped with cellSens imaging software (version 1.12). IODsum was calculated for arbitrary areas (20 arbitrary areas except for necrotic areas, 900 μm × 500 μm) and the average was calculated. Each section was analyzed with the same size. We defined low expression of HIF-1α or GLUT1 as IODsum value ≤ the median, and high expression as IODsum value > the median.

Hematoxylin-eosin staining: The surgical specimens were fixed in 10% buffered formalin and serially cut into 5-mm-thick sections, then stained with hematoxylin-eosin. The following pathological features were assessed: Histologic grading of PDACs was classified according to Adsay et al. [14]. PDACs in well-differentiated (grade I) and moderately differentiated (grade II) tumors were classified as low-grade tumors; PDACs in poorly differentiated (grade III) tumors were classified as high-grade tumors. Tumor cellularity was classified as <50% or ≥50%; Extent of fibrosis was classified as mild (33%), moderate (33–66%) or severe (33–66%) according to the ratio of fibrosis in the tumor [15]; we also assessed the remaining acini and classified them as present or absent.

The assessment of histopathologic lymph node status was performed according to the American Joint Committee on Cancer (AJCC, 7th edition). N0 was defined as no regional lymph node involvement, and N1 was defined as regional lymph node involvement.

The inflammatory scores were calculated as follows [16]: NLR (neutrophil-lymphocyte ratio) = absolute neutrophil count (ANC)/absolute lymphocyte count (ALC); PLR (platelet-to-lymphocyte ratio) = absolute platelet count (APC)/ALC; Systemic immune-inflammation index (SII) = (ANC × APC)/ALC.

### 2.4. Statistical Analysis

Pearson’s chi-square test or Fisher’s exact test was used to compare frequencies of categorical variables. Continuous variables were evaluated by using Student’s *t* test if the assumption of normality was satisfied; otherwise, the Mann–Whitney U test was used. Multivariate logistic regression analysis was performed based on the significant variables acquired from univariate analysis. Pearson’s correlation analysis was performed to examine the correlation between the IOD in HIF-1α and GLUT1 staining. Sensitivity, specificity and accuracy calculations were performed among independent predictors identified in the multivariable model. The receiver operating characteristic (ROC) analysis was used to identify optimal cutoff values of the continuous variables and evaluate AUC, sensitivity and specificity. Statistical analysis was performed using software (SPSS, version 26.0; GraphPad Prism, version 9.5.0, La Jolla, CA, USA). A *p* < 0.05 was defined as statistically significant. 

## 3. Results

### 3.1. Correlations between Clinicopathological Features and the Expression of HIF-1α and GLUT1 of PDAC

The mean IODsum in HIF-1α and GLUT1 staining was 29,918.1 (range, 1444–147,818; median 23,955) and 29,438.6 (range, 924–149,483; median 20,661), respectively. Both HIF-1α-high PDACs and GLUT1-high PDACs had a larger tumor size (3.68 ± 1.76 vs. 3.07 ± 1.03 *p* = 0.002; 3.80 ± 1.70 vs. 2.90 ± 1.10 *p* = 0.002, respectively) and were more prone to lymph node metastasis (*p* = 0.009 and *p* = 0.026, respectively). PDACs with severe fibrosis (>66%) had a tendency to be HIF-1α-high and have GLUT1 high expression. Among the PDAC with severe fibrosis, 10 (66.7%, 10/15) cases showed HIF-1α high expression and 11 (73.3%, 11/15) cases showed GLUT1 high expression, but the difference was not statistically significant (*p* = 0.228, and *p* = 0.140, respectively) (Table 1).

### 3.2. Correlation between the Expression of HIF-1α and GLUT1 in PDAC

Among 102 PDAC tissues, integrated optical density (IOD) was used to quantitatively analyze the immunostaining intensity. A positive correlation was found between the expression of HIF-1α and GLUT1 (r = 0.490, *p* < 0.0001) (Figure 1).

### 3.3. Correlations between CEUS Qualitative Characteristics and the Expression of HIF-1α and GLUT1 of PDAC

HIF-1α-high PDACs (*p* = 0.001) and GLUT1-high PDACs (*p* = 0.003) manifested incomplete enhancement more frequently on CEUS (Figure 2 and Figure 3). In PDACs with ≤50% enhancement, 80% (12/15) showed HIF-1α high expression and GLUT1 high expression. The peak enhancement degree (PED) in HIF-1α-high PDACs (*p* = 0.027) and GLUT1-high PDACs (*p* = 0.008) showed iso-enhancement more frequently (Figure 2 and Figure 3) (Table 2).

### 3.4. Model Development for Predicting HIF-1α and GLUT1 Expression Levels

A univariate logistic regression analysis showed significant differences in tumor size, lymph node metastasis, completeness of enhancement and PED. The multivariate logistic regression analysis revealed that the tumor size, lymph node metastasis, incomplete enhancement and isoenhancement of PED were all independent predictors for both HIF-1 α-high PDACs and GLUT1-high PDACs (Table 3) 

### 3.5. Diagnostic Performance

Table 4 shows the diagnostic performance for HIF-1 α-high PDACs and GLUT1-high PDACs for the independent predictors identified in the multivariable model. The sensitivity, specificity and accuracy for lymph node metastasis in predicting HIF-1 α-high PDACs and GLUT1-high PDACs were 52.9%, 72.5%, 62.7% and 51.0%, 70.6%, 60.8%, respectively; for incomplete enhancement, 66.7%, 68.6%, 67.6% and 64.7%, 66.7%, 65.7%, respectively; and for isoenhancement of PED, 37.3%, 82.4%, 59.8% and 39.2%, 84.3%, 61.8%, respectively. By using the ROC area under curve (AUC) analysis, the optimal cutoff value for tumor size in predicting HIF-1 α-high PDACs was found to be 3.35 (AUC = 0.700, 95% CI 0.60–0.80, 80.4% specificity, and 57% sensitivity) and in predicting GLUT1-high PDACs, it was 3.45 (AUC = 0.715, 95% CI 0.62–0.82, 82.4% specificity, and 51% sensitivity) (Figure 4).

### 3.6. Correlations between CEUS Quantitative Perfusion Features and the Expression of HIF-1α and GLUT1 of PDAC

A quantitative analysis of CEUS was performed using the Software VueBox Quantification Toolbox (version 4.3). A total of 72 of 102 lesions obtained the ROIs parameters. Among the 72 cases of PDAC, 24 cases showed both high HIF-1α and GLUT1 high expression, 21 cases showed both low HIF-1α and GLUT1 low expression, 12 cases showed high HIF-1α but GLUT1 low expression, and 15 cases showed low HIF-1α but GLUT1 high expression. Both HIF-1α-high PDACs and GLUT1-high PDACs showed higher peak enhancement (PE) (*p* = 0.024 and *p* = 0.018, respectively) (Figure 5 and Figure 6) and higher wash-in rate (WIR) values (*p* = 0.015 and 0.002, respectively) (Table 5) more frequently. There were no significant differences in rise time (RT), time to peak (TTP) and wash-in area under the curve (WiAUC) between HIF-1α or GLUT1 high and low expression PDACs (*p* > 0.05 for all).

## 4. Discussion

CEUS is exquisitely sensitive in depicting the microvascularity of tissues due to its blood pool nature [5,17]. Recently, CEUS has been used as a non-invasive tool for assessing the tumor microenvironment. Jia et al. [18] have showed that CEUS features can help evaluate tumor-infiltrating lymphocytes (TILs) in breast cancer. Shah et al. [19] have demonstrated that CEUS features were significantly correlated with the tumor’s blood oxygen saturation and hemoglobin, which can be used as imaging biomarkers of hypoxia. Our study explored the association between CEUS characteristics and the hypoxic microenvironment of PDAC. We demonstrated that tumor size, lymph node metastasis, incomplete enhancement and iso-enhancement of PED were independent predictors for HIF-1 α-high PDACs and GLUT1-high PDACs, and CEUS quantitative perfusion parameters, including PE and WIR values, were higher in HIF-1α-high PDACs and GLUT1-high PDACs.

PDAC exhibits higher levels of hypoxia than most solid tumors. The presence of intratumoral hypoxia is strongly associated with tumor biological behaviors or malignant phenotypes such as cancer proliferation, metastasis, therapeutic resistance and angiogenesis [1,3,20]. The adaptive responses to a hypoxic microenvironment stimulate a more aggressive and treatment-refractory phenotype of PDAC, and hypoxia may be the key point to improving current treatment strategies. [3,21]. Innovative therapies which target the tumor hypoxic hold great potential to overcome the chemoresistance and radioresistance and, thus, enhance the therapy’s efficacy [3]. It has been reported that a high expression of HIF-1α in PDAC markedly reduced sensitivity to gemcitabine [22]. HIF-1α inhibitors can be used in HIF-1α high expression PDACs to improve therapeutic efficacy by inhibiting the overexpression of HIF-1α or promoting the degradation of HIF-1α [23,24]. Zhang et al. [25] have reported that inhibition of HIF-1α enhances the sensitivity to gemcitabine in PDAC via suppression of autophagic fux. Shukla et al. [26] have also suggested that targeting HIF-1α or HIF-1α-mediated metabolic pathways enhances the efficacy of gemcitabine. GLUT1 is a high-affinity glucose transporter that regulates glucose uptake with increased expression during hypoxia [27]. Glucomet-PDACs (PDAC with high glucose metabolism levels) have been reported to be resistant to chemotherapy, and pharmacological inhibition of GLUT1 enhances the chemotherapy response of glucomet-PDAC [28]. With a better understanding of the expression levels of HIF-1α and GLUT1, more effective and patient-specific therapies could be devised. In addition, HIF-1α expression is associated with the angiogenesis, cell proliferation and metastasis of pancreatic cancer [29,30]. Sun et al. [9] have shown that higher HIF-1α and GLUT1 expression indicated lymph node metastasis and a tendency of a larger tumor size, which were similar to our results. Carbohydrate antigen 19-9 (CA 19-9) is an important tumor marker for the diagnosis, management and prognosis prediction of PDAC [31]. Imamura et al. [32] have proven that elevated CA19-9 was one of the independent predictors for the early recurrence of PDAC after pancreatectomy. Kanda et al. [33] have reported that high CA19-9 PDAC demonstrated aerobic glycolysis enhancement. Wang et al. [34] have reported that the elevated serum CA19-9 is associated with a single nucleotide polymorphism (SNP) of the HIF-1α gene in PDAC, which is a DNA sequence variation and may be involved in oncogene expression. We tried to analyze the relationship between CA19-9 level and the HIF-1α/GLUT1 expression of PDAC, but no significant differences were found between groups. Similarly, some blood indices such as systemic immune-inflammation index (SII), neutrophil–lymphocyte ratio (NLR) and platelet-to-lymphocyte ratio (PLR) have been proven to be important prognostic indicators for PDAC [35,36]. Jomrich et al. [36] have suggested that PDACs with elevated SII might benefit from anti-inflammatory or anti-immunotherapy. Murthy et al. [37] have reported that SII could be used as a biomarker of response to neoadjuvant therapy in patients with PDAC. In this study, no significant differences were found between these blood indices and tumor hypoxia.

Our study established CEUS imaging as a reliable and valuable method for measuring the tumor hypoxic microenvironment of PDACs. Incomplete enhancement and iso-enhancement of PED were the most statistically significant parameters in predicting HIF-1α and GLUT1 levels. Prolonged hypoxia of the tumor tissue leads to necrosis. Hypoxic tumor cells undergo apoptosis and become necrotic and coalesce to form the necrotic core of larger tumors [38]. When necrotic areas are formed, the tumors show incomplete enhancement on CEUS. The present study showed that incomplete enhancement was one of the independent predictors for HIF-1 α-high PDACs and GLUT1-high PDACs, and among PDACs with enhancement ≤50%, 80% (12/15) showed HIF-1α high expression and GLUT1 high expression. Our study demonstrated that both HIF-1 α-high PDACs and GLUT1-high PDACs more frequently showed iso-enhancement of PED and had higher PE values as well. This may be due to the role of the tumor hypoxic microenvironment in promoting tumor angiogenesis. Previous studies [9,39] have reported that the expression of HIF-1α and GLUT1 correlated positively with the vascular endothelial growth factor (VEGF) and microvessel density (MVD) in various solid tumors. Notably, for the lesions which showed hypoenhancement on CEUS, other potential causes might be intratumoral necrosis, severe fibrosis or mucin [40]. Endoscopic ultrasound (EUS) is an important diagnostic technique for pancreatic lesions and allows tissue sampling by means of fine-needle aspiration (FNA) and fine-needle biopsy (FNB) [41]. CH-EUS (contrast-enhanced harmonic endoscopic ultrasound) makes it possible to combine EUS with contrast agent injection, achieving precise imaging of the interior of the pancreas. Endoscopic ultrasound (EUS) contrast-enhanced fine-needle aspiration (CH-EUS-FNA) has been reported to be superior to EUS-FNA in determining the ideal target area, because it can help avoid necrotic areas and vessels within the tumor. [42]. The results of this study can be verified by CH-EUS and CH-EUS-FNA, owing to its superiority that allows detailed visualization of the intratumor blood flow of PDACs, precise sampling of target enhancement tissues and point-to-point analysis of enhancement characteristics and tumor hypoxia status. In addition, our study can help to establish a method to verify the amount and exact locations of hypoxic microenvironments of PDAC with CEUS or CH-EUS information during tissue sampling. With targeted tissue sampling, hypoxic and potentially chemoresistant cancer tissue can be precisely obtained for further revealing hypoxia-related alterations and identifying key molecules of hypoxic tumors.

Currently, the hypoxic imaging assessment has become a research hotspot because it would be an integral biomarker in defining tumor progression and treatment resistance. Dynamic contrast-enhanced magnetic resonance imaging (DCE-MRI) has been reported to have great values in assessing the microenvironment of PDAC and may adequately reflect tumor vascular density and hypoxia in treated PDACs [43,44]. Klaassen et al. [45] evaluated the reproducibility and interaction of DCE-MRI parameters transfer constant (K^trans^), rate constant (k_ep_), extracellular extravascular space (v_e_) and perfusion fraction (v_p_), as well the T2* MRI parameter R2* (1/T2*) maps based on 15 PDAC patients; they found that DCE- and T2*-related MRI parameters were well reproducible and have substantial value in cumulatively capturing tumor perfusion and hypoxia in patients with advanced PDAC. However, no comparison with tissue hypoxia markers was performed to demonstrate whether these parameters hold promise as imaging biomarkers of hypoxia in PDAC. Then, Klaassen et al. [46] have further proved that MRI is able to accurately characterize tumor hypoxia in PDAC via analyzing the correlation between MRI parameters and immunohistochemistry-derived tissue characteristics. There have also been several ongoing clinical studies on the development of imaging biomarkers for tumor hypoxic microenvironments. The imaging modalities involved mainly included MRI (NCT05029258, NCT05904704) and positron emission tomography (PET) (NCT03054792), and the tumor types involved mainly included cervical cancer, intracranial cancers and sarcomas. One ongoing clinical study (NCT04395469) aimed to assess whether PET-MRI scans can provide useful information about hypoxia in pancreatic cancer. However, there were no ongoing clinical studies which aimed to assess hypoxia in pancreatic cancer via CEUS imaging. Our study proved that CEUS, which was fast, safe, repeatable and relatively economic, was of great significance in reflecting the hypoxic microenvironment of PDAC, and large-scale clinical trials in this regard could be carried out in the future.

There were some limitations to this study. Firstly, there might have been a potential selection bias because of the retrospective nature. Secondly, our results may not allow definite conclusions for all PDACs because only PDAC with preoperative CEUS and surgical excision were included. Thirdly, CEUS is a highly operator-dependent imaging modality, needing specific training and equipment, and the imaging results may be suboptimal due to patient-related factors such as a high BMI, bowel gas or ascites. In addition, pancreatic body or tail lesions are sometimes difficult to identify via transabdominal CEUS. The results of this study should be verified by CH-EUS and CH-EUS-FNA in the future. Lastly, this was a single-centered study with a limited number of patients, and quantitative analysis parameters were not obtained for all lesions. Multicenter prospective studies with a larger number of patients are needed to confirm our findings.

## 5. Conclusions

In conclusion, CEUS characteristics can non-invasively reflect the hypoxic microenvironment of PDAC. Our study provided a reliable imaging biomarker for tumor hypoxia that could be easily implemented into a routine clinical workflow, which can not only identify potentially chemoresistant cancer but also evaluate the efficacy of hypoxia-targeted therapies. In addition, because tumor hypoxia is an adverse feature related to reduced survival, CEUS-based tumor hypoxia evaluation could provide prognostic information for patients with PDAC.

## Figures and Tables

**Figure 1 diagnostics-13-03270-f001:**
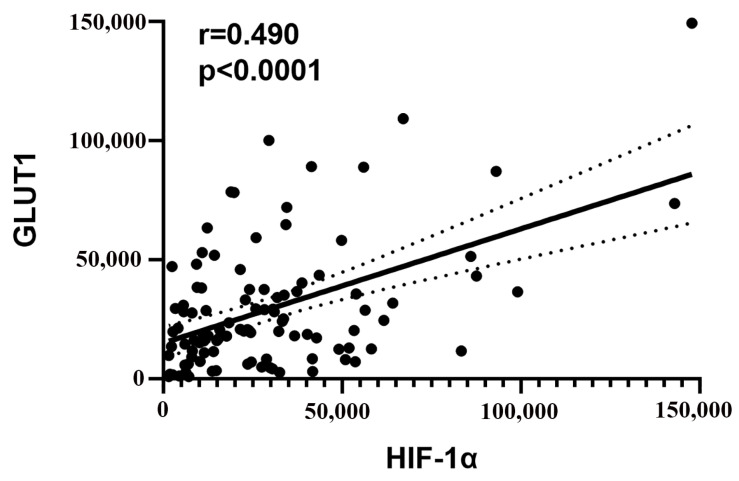
Correlation analysis between the expression of HIF-1α and GLUT1. A positive correlation was found between the expression (IODsum) of HIF-1α and GLUT1 of PDAC.

**Figure 2 diagnostics-13-03270-f002:**
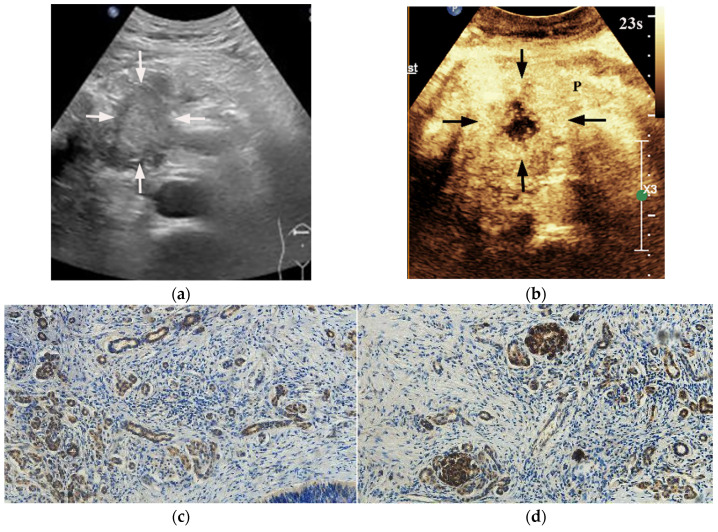
A 29-year-old male patient with a pancreatic ductal adenocarcinoma. (**a**) US revealed a hypoechoic lesion (arrows) at the pancreatic head. (**b**) The lesion (arrows) demonstrated incomplete enhancement (enhancement > 50%) on CEUS and the peak enhancement degree of the solid component was iso-enhancement. P = normal pancreatic parenchyma. (**c**) Photomicrographs of immunohistochemistry stain sections demonstrate a HIF-1α high expression ductal adenocarcinoma (original magnification, ×200). (**d**) Photomicrographs of immunohistochemistry stain sections demonstrate a GLUT1 high expression ductal adenocarcinoma (original magnification, ×200).

**Figure 3 diagnostics-13-03270-f003:**
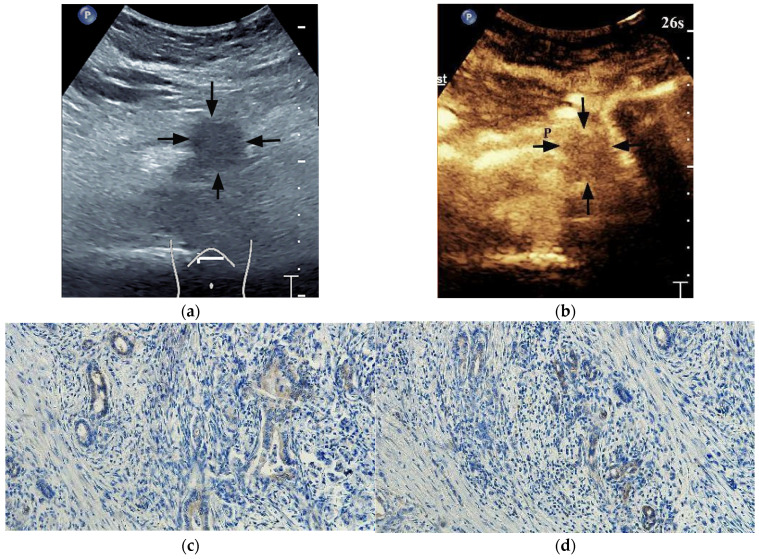
A 48-year-old male patient with a pancreatic ductal adenocarcinoma. (**a**) US revealed a hypoechoic lesion (arrows) at the pancreatic tail. (**b**) The lesion (arrows) demonstrated complete enhancement and hypo-enhancement on CEUS. P = normal pancreatic parenchyma. (**c**) Photomicrographs of immunohistochemistry stain sections demonstrate a HIF-1α low expression ductal adenocarcinoma (original magnification, ×200). (**d**) Photomicrographs of immunohistochemistry stain sections demonstrate a GLUT1 low expression ductal adenocarcinoma (original magnification, ×200).

**Figure 4 diagnostics-13-03270-f004:**
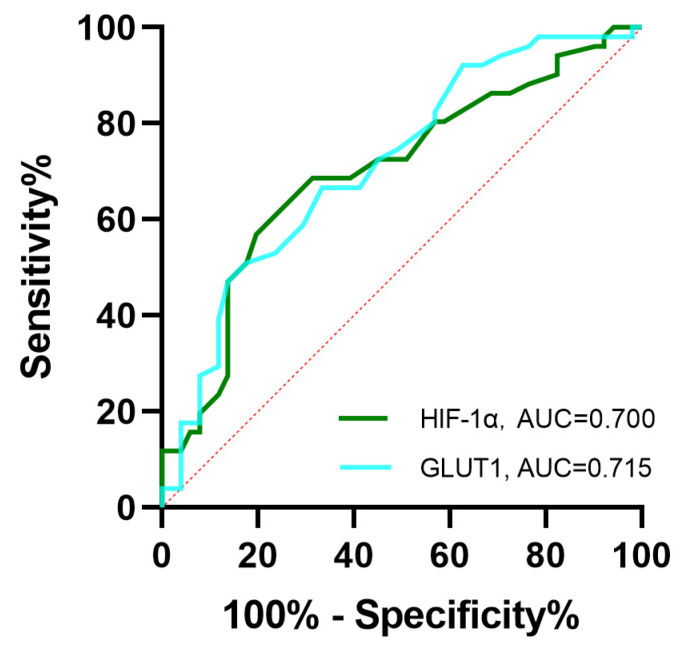
Receiver operating characteristic (ROC) curve of tumor size to predict the level of HIF-1α (green curve) and GLUT1 (blue curve) in PDAC. An AUC of 0.700 with 80.4% specificity and 57% sensitivity and optimal cutoff value for tumor size in predicting low and high HIF-1α levels was 3.35 cm; an AUC of 0.715 with 82.4% specificity and 51% sensitivity and optimal cutoff value for tumor size in predicting low and high GLUT1 levels was 3.45 cm.

**Figure 5 diagnostics-13-03270-f005:**
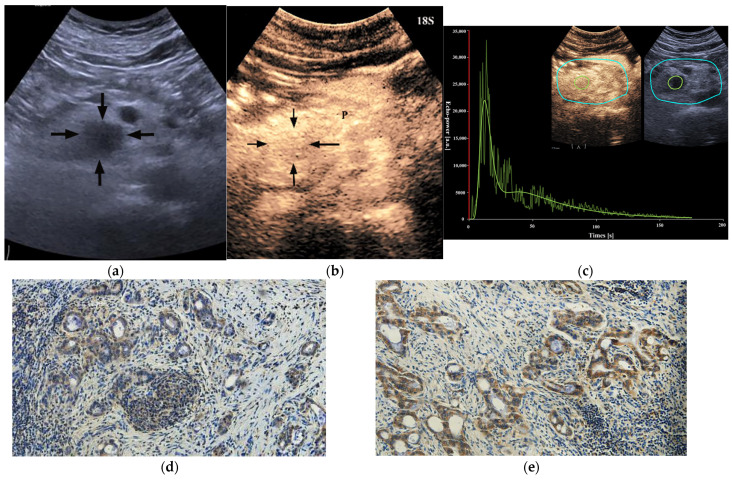
A 60-year-old male patient with a pancreatic ductal adenocarcinoma. (**a**) US revealed a hypoechoic lesion (arrows) at the pancreatic head. (**b**) The lesion (arrows) showed iso-enhancement of PED on CEUS. P = normal pancreatic parenchyma. (**c**) Time intensity curve analysis demonstrated a higher peak enhancement value (21,946.5 [a.u.]). (**d**) Photomicrographs of immunohistochemistry stain sections demonstrated a HIF-1α high expression ductal adenocarcinoma (original magnification, ×200). (**e**) Photomicrographs of immunohistochemistry stain sections demonstrated a GLUT1 high expression ductal adenocarcinoma (original magnification, ×200).

**Figure 6 diagnostics-13-03270-f006:**
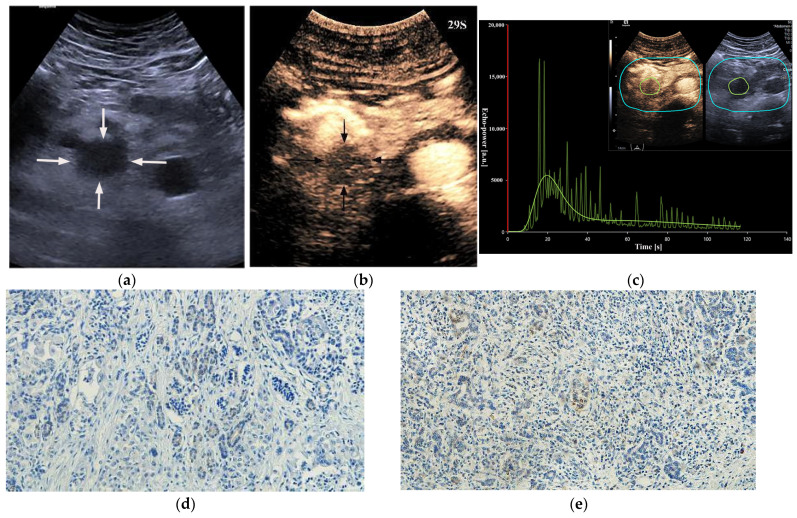
A 69-year-old male patient with a pancreatic ductal adenocarcinoma. (**a**) US revealed a hypoechoic lesion (arrows) at the pancreatic head. (**b**) The lesion (arrows) demonstrated hypo-enhancement on CEUS. (**c**) Time intensity curve analysis demonstrated a relatively lower peak enhancement value (5420 [a.u.]). (**d**) Photomicrographs of immunohistochemistry stain sections demonstrate a HIF-1α low expression ductal adenocarcinoma (original magnification, ×200). (**e**) Photomicrographs of immunohistochemistry stain sections demonstrate a GLUT1 low expression ductal adenocarcinoma (original magnification, ×200).

**Table 1 diagnostics-13-03270-t001:** Relationship between clinicopathological features and the expression of HIF-1α and GLUT1 of PDAC.

	HIF-1α			GLUT1		
	Low (*n* = 51)	High (*n* = 51)	*p* Value	Low (*n* = 51)	High (*n* = 51)	*p* Value
Age, y	60.18 ± 8.11	58.82 ± 8.51	0.413	61.08 ± 7.83	57.92 ± 8.53	0.054
Location						
Head	40 (78.4%)	34 (66.7%)	0.183	37 (72.5%)	37 (72.5%)	1.00
Body or tail	11 (21.6%)	17 (33.3%)		14 (27.5%)	14 (27.5%)	
Tumor size, cm	3.07 ± 1.03	3.68 ± 1.76	0.002	2.90 ± 1.10	3.80 ± 1.70	0.002
Tumor grade						
Low grade	33 (64.7%)	28 (54.9%)	0.313	34 (66.7%)	27 (52.9%)	0.157
High grade	18 (35.3%)	23 (45.1%)		17 (33.3%)	24 (47.1%)	
Tumor cellularity						
≥50%	25 (49.0%)	33 (64.7%)	0.110	26 (51.0%)	32 (60.4%)	0.230
<50%	26 (51.0%)	18 (35.3%)		25 (49.0%)	19 (39.6%)	
Remaining acini						
Presence	17 (33.3%)	22 (43.1%)	0.308	23 (45.1%)	16 (31.4%)	0.154
Absence	34 (66.7%)	29 (56.9%)		28 (54.9%)	35 (68.6%)	
Fibrosis extent						
Mild fibrosis (<33%)	26 (51.0%)	18 (35.3%)	0.228	23 (45.1%)	21 (41.2%)	0.140
Moderate fibrosis (33–66%)	20 (39.2%)	23 (45.1%)		24 (47.1%)	19 (37.3%)	
Severe fibrosis (>66%)	5 (9.8%)	10 (19.6%)		4 (7.8%)	11 (21.5%)	
Lymph node metastasis						
N1	14 (27.5%)	27 (52.9%)	0.009	15 (29.4%)	26 (51.0%)	0.026
N0	37 (72.5%)	24 (47.1%)		36 (70.6%)	25 (49.0%)	
Resection margin status						
R0	47 (92.2%)	45 (88.2%)	0.505	46 (90.2%)	46 (90.2%)	1.00
R1	4 (7.8%)	6 (11.8%)		5 (9.8%)	5 (9.8%)	
Bile duct dilatation						
Yes	29 (56.9%)	30 (58.8%)	0.841	28 (54.9%)	31 (60.8%)	0.547
No	22 (43.1%)	21 (41.2%)		23 (45.1%)	20 (39.2%)	
Pancreatic duct dilation						
Yes	31 (60.8%)	32 (60.4%)	0.839	35 (68.6%)	28 (54.9%)	0.154
No	20 (39.2%)	19 (39.6%)		16 (31.4%)	23 (45.1%)	
Serum CA 19-9						
≥200 U/mL	14 (27.5%)	22 (43.1%)	0.097	16 (31.4%)	20 (39.2%)	0.407
<200 U/mL	37 (72.5%)	29 (56.9%)		35 (68.6%)	31 (60.8%)	
SII	582.96 (334.61–965.64)	504.00 (369.19–810.93)	0.428	511.31 (349.32–830.07)	582.96 (359.70–823.32)	0.838
NLR	3.35 (2.23–4.79)	2.91 (2.15–3.99)	0.304	2.86 (2.05–4.0)	3.24 (2.45–4.57)	0.505
PLR	158.76 (93.65–229.27)	151.16 (112.57–193.90)	0.599	152.24 (95.60–223.53)	156.38 (100.80–213.75)	0.880

HIF-1α = Hypoxia-inducible factor-1α, GLUT1 = Glucose transporter-1, SII = Systemic inflammatory index, NLR = Neutrophil-to-lymphocyte ratio, PLR = Platelets-to-lymphocyte ratio.

**Table 2 diagnostics-13-03270-t002:** Relationship between CEUS qualitative characteristics and the expression of HIF-1α and GLUT1 of PDAC.

CEUS	HIF-1α			GLUT1		
	Low (*n* = 51)	High (*n* = 51)	*p* Value	Low (*n* = 51)	High (*n* = 51)	*p* Value
Wash-in time						
Late	38 (74.5%)	29 (56.9%)	0.061	33 (64.7%)	34 (66.7%)	0.835
Early/Synchronous	13 (25.5%)	22 (43.1%)		18 (35.3%)	17 (33.3%)	
Centripetal enhancement						
Yes	22 (43.1%)	29 (56.9%)	0.166	23 (45.1%)	28 (54.9%)	0.322
No	29 (56.9%)	22 (43.1%)		28 (54.9%)	23 (45.1%)	
Penetrating vessels						
Presence	19 (37.3%)	23 (45.1%)	0.421	18 (35.3%)	24 (47.1%)	0.227
Absence	32 (62.7%)	28 (54.9%)		33 (64.7%)	27 (52.9%)	
Completeness of enhancement						
Complete enhancement	35 (68.6%)	17 (33.3%)	0.001	34 (66.7%)	18 (35.3%)	0.003
Incomplete enhancement (enhancement > 50%)	13 (25.5%)	22 (43.1%)		14 (27.5%)	21 (41.2%)	
Incomplete enhancement (enhancement ≤ 50%)	3 (5.9%)	12 (23.5%)		3 (5.8%)	12 (23.5%)	
PED						
Iso-enhancement	9 (17.6%)	19 (37.3%)	0.027	8 (15.7%)	20 (39.2%)	0.008
Hypo-enhancement	42 (82.4%)	32 (62.7%)		43 (84.3%)	31 (60.8%)	
Wash-out time						
Early	36 (70.6%)	38 (74.5%)	0.657	34 (66.7%)	40 (78.4%)	0.183
Late/Synchronous	15 (29.4%)	13 (25.5%)		17 (33.3%)	11 (21.6%)	

HIF-1α = Hypoxia-inducible factor-1α, GLUT1 = Glucose transporter-1, PED = peak enhancement degree.

**Table 3 diagnostics-13-03270-t003:** Univariate and multivariate logistic regression analyses of clinicopathological features and CEUS features predictive of HIF-1α and GLUT1 levels.

	HIF-1α				GLUT1			
	Univariable		Multivariable		Univariable		Multivariable	
	Odds Ratio (95% CI)	*p*	Odds Ratio (95% CI)	*p*	Odds Ratio (95% CI)	*p*	Odds Ratio (95% CI)	*p*
Clinicopathological features								
Age	0.98(0.94–1.03)	0.410			0.95(0.91–1.00)	0.059		
Location	1.82(0.75–4.41)	0.186			1.00(0.42–2.39)	1.00		
Tumor size	1.81(1.20–2.73)	0.005 *	1.54(1.01–2.34)	0.044	1.83(1.21–2.78)	0.004 *	1.59(1.04–2.43)	0.031
Tumor grade	1.51(0.68–3.34)	0.314			1.78(0.80–3.96)	0.159		
Tumor cellularity	1.91(0.86–4.22)	0.111			1.62(0.74–3.57)	0.232		
Remaining acini	0.66(0.30–1.47)	0.309			1.90(0.80–4.03)	0.156		
Severe fibrosis	2.24(0.71–7.11)	0.170			3.23(0.95–10.94)	0.059		
Lymph node metastasis	2.97(1.30–6.78)	0.10 *	3.36(1.32–8.59)	0.011	2.50(1.11–5.64)	0.028 *	2.73(1.08–6.86)	0.033
Resection margin status	1.57(0.42–5.92)	0.508			1.00(0.27–3.69)	1.00		
Bile duct dilatation	1.08(0.49–2.38)	0.841			1.27(0.58–2.80)	0.548		
Pancreatic duct dilation	1.09(0.49–2.42)	0.839			0.56(0.25–1.25)	0.156		
Serum CA 19-9	0.50(0.22–1.14)	0.100			0.71(0.31–1.60)	0.408		
SII	0.99(0.999–1.000)	0.069			1.000(0.999–1.000)	0.342		
NLR	0.88(0.75–1.02)	0.092			0.95(0.85–1.06)	0.324		
PLR	0.998(0.994–1.003)	0.502			0.999(0.995–1.004)	0.734		
CEUS features								
Wash-in time	2.22(0.96–5.13)	0.063			0.917(0.41–2.08)	0.835		
Centripetal enhancement	0.58(0.26–1.26)	0.167			0.68(0.31–1.47)	0.323		
Penetrating vessels	1.38(0.63–3.05)	0.422			1.63(0.74–3.61)	0.229		
Completeness of enhancement	4.38(1.91–10.03)	<0.001 *	3.39(1.33–8.64)	0.011	3.67(1.62–8.31)	0.002 *	2.71(1.07–6.86)	0.035
Peak enhancement degree	2.77(1.11–6.93)	0.029 *	3.08(1.08–8.80)	0.035	3.47(1.35–8.89)	0.01 *	3.88(1.36–11.07)	0.011
Wash-out time	0.82(0.34–1.96)	0.657			0.55(0.23–1.33)	0.186		

* Significant at univariate analysis and carried onward to multivariate analysis, CI = confidence interval, HIF-1α = Hypoxia-inducible factor-1α, GLUT1 = Glucose transporter-1, SII = Systemic inflammatory index, NLR = Neutrophil-to-lymphocyte ratio, PLR = Platelets-to-lymphocyte ratio.

**Table 4 diagnostics-13-03270-t004:** Performance of optimal features for predicting HIF-1α and GLUT1 levels.

	HIF-1α			GLUT1		
	Sensitivity	Specificity	Accuracy	Sensitivity	Specificity	Accuracy
Lymph node metastasis	52.9%	72.5%	62.7%	51.0%	70.6%	60.8%
Incomplete enhancement	66.7%	68.6%	67.6%	64.7%	66.7%	65.7%
Isoenhancement of PED	37.3%	82.4%	59.8%	39.2%	84.3%	61.8%

HIF-1α = Hypoxia-inducible factor-1α, GLUT1 = Glucose transporter-1, PED = peak enhancement degree.

**Table 5 diagnostics-13-03270-t005:** Relationship between CEUS quantitative features and the expression of HIF-1α and GLUT1 of PDAC.

Parameter, Median (IQR)	HIF-1α			GLUT1		
	Low (*n* = 36)	High (*n* = 36)	*p* Value	Low (*n* = 33)	High (*n* = 39)	*p* Value
RT [s]	8.0 (6.7–10.4)	7.0 (6.2–8.7)	0.154	8.0 (6.7–12.1)	7.2 (6.0–9.7)	0.067
TTP [s]	12.1 (9.7–16.0)	11.3 (8.8–14.5)	0.265	12.8 (9.8–15.9)	11.2 (8.7–13.3)	0.076
PE [a.u.]	10,693.4 (6566.8–20,600.4)	16,173.8 (9381.0–27,237.6)	0.024	10,125.9 (7037.8–18,662.1)	15,939.4 (8939.2–27,388.2)	0.018
WIR [a.u.]	1830.6 (868.4–4287.2)	3607.1 (1973.7–5895.6)	0.015	1898.3 (913.5–3844.3)	3882.5 (1967.1–6205.7)	0.002
WiAUC [a.u.]	57,775.0 (31,068.1–98,571.4)	80,406.4 (45,124.3–123,632.3)	0.169	57,933.7 (35,078.5–98,126.3)	78,038.4 (43,797.8–133,500.0)	0.334

Median (IQR) = Median (Interquartile range), HIF-1α = Hypoxia-inducible factor-1α, GLUT1 = Glucose transporter-1, RT = rise time, TTP = Time to peak, PE = Peak enhancement, WIR = Wash-in rate, WiAUC = Wash-in Area Under the Curve.

## Data Availability

The data that support the findings of this study are available from the corresponding author upon reasonable request.

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
