# Peer review of "Pancreatic Ductal Adenocarcinoma: The Characteristics of Contrast-Enhanced Ultrasound Are Correlated with the Hypoxic Microenvironment"

_diagnostics, 2023, doi:10.3390/diagnostics13203270_

Round 1

Reviewer 1 Report

This paper describes the relationship between CE-EUS characteristics and HIF-1/GLUT-1). It is interesting and worth to read for clicians and researchers. Hypoxic microeviroment has been reported to be associated with the progression of PDAC. 

The authors are expected to show the histopathological finding more in detail, such as defferentiation and cancer-stroma relationship.

Author Response

For research article

Response to Reviewer 1 Comments

1. Summary

Thank you very much for taking the time to review this manuscript. Please find the detailed responses below and the corresponding revisions in the re-submitted files. All corrections are highlighted by using the 'track changes' option in Microsoft Word. In any case, we are open to consideration of any further comments and suggestions.

2. Questions for General Evaluation

Reviewer’s Evaluation

Response and Revisions

Does the introduction provide sufficient background and include all relevant references?

Yes

Thanks for your evaluation.

Are all the cited references relevant to the research?

Yes

Thanks for your evaluation.

Is the research design appropriate?

Yes

Thanks for your evaluation.

Are the methods adequately described?

Yes

Thanks for your evaluation.

Are the results clearly presented?

Looking forward to your evaluation.

Are the conclusions supported by the results?

Yes

Thanks for your evaluation.

3. Point-by-point response to Comments and Suggestions for Authors

Comments 1: This paper describes the relationship between CE-EUS characteristics and HIF-1/GLUT-1). It is interesting and worth to read for clicians and researchers. Hypoxic microeviroment has been reported to be associated with the progression of PDAC.

The authors are expected to show the histopathological finding more in detail, such as differentiation and cancer-stroma relationship.

Response 1: Thank you for your valuable suggestions to improve the quality of our manuscript. We have supplemented the histopathological finding more in detail, and we have further compared the relationship between the histopathological finding (including tumor cellularity, remaining acini, and fibrosis extent) and hypoxic microenvironmental (the expression of HIF-1α and GLUT1) of PDAC. Our study found that PDACs with severe fibrosis (>66%) had a tendency to be HIF-1α-high  and  GLUT1-high  expression. Among the PDAC with severe fibrosis (>66%), 10 (66.7%, 10/15) cases showed HIF-1α-high expression and 11 (73.3%, 11/15) cases showed GLUT1-high expression, but the difference had no statistically significant (p=0.228, and p=0.140, respectively). These additions can be found in Table 1, Table 3, and in the text on page 4, line 174-182 and on page 5, line 211-215. As for the differentiation of PDAC, I should have explained that we have analyzed the relationship between the histologic grade and hypoxic microenvironment of PDAC in our original manuscript. According to Adsay et al. [Reference 14 in the text], PDACs in well-differentiated (grade I) and moderately differentiated (grade II) were classified as low-grade tumors, and PDACs in poorly differentiated (grade III) tumors were classified as high-grade tumors. No significant correlation between pathological grade and hypoxic microenvironment was found in this study (p>0.05). The contents can be found in Table 1, Table 3 and in the text on page 4, line 176-179.

Sincerely,

Fang Nie

Reviewer 2 Report

I read with great interest the novel retrospective research study showing CEUS predicting HIF-alpha and GLUT-1 in PDAC patients. This study complements the existing knowledge and has potential to prognosticate PDAC patients in the preoperative phase and such information may be crucial in treatment plans and recommendations for upfront resection versus neoadjuvant chemotherapy.

I have minor comments.

1, Table 5 why does HIF alpha have 36 - 36 groups but GLUT-1 has 33 and 39 groups?

2, You should discuss the drawback of CEUS being operator-dependent, needing special training and facilities/resources.

3, You should discuss the clinical implications of knowing that a PDAC patient has high HIF-alpha and GLUT-1. How does the care/treatment differ compared to not knowing this information? I do not see any discussion on the clinical application of this research.

4, As CEUS is not a good modality to image pancreas, is it possible that EUS can be done with SonoVue injection and pancreas imaged internally and if that would enhance the accuracy? or that would help identify areas to perform Needle biopsy? Anyone has done this? Some discussion is warranted.

5, Authors must mention some data about CA19-9 which is a tumour marker relevant in PDAC - PMID: 33437400

6, I do see authors mentioning some blood indices / ratios but there is minimal discussion about the utility of such indices in PDAC.

7, Authors have 100 over patient data about whipples / PDAC surgery and some operative details is necessary - how many whipples and how many distal panc, 30 and 90 day mortality, POPF, DGE, PPH and other relevant demographic details of the patients should be mentioned in tables

Thanks

Author Response

For research article

Response to Reviewer 2 Comments

1. Summary

Thank you very much for taking the time to review this manuscript. Please find the detailed responses below and the corresponding revisions in the re-submitted files. All corrections are highlighted by using the 'track changes' option in Microsoft Word. In any case, we are open to consideration of any further comments and suggestions.

2. Questions for General Evaluation

Reviewer’s Evaluation

Response and Revisions

Does the introduction provide sufficient background and include all relevant references?

Yes

Thanks for your evaluation.

Are all the cited references relevant to the research?

Yes

Thanks for your evaluation.

Is the research design appropriate?

Yes

Thanks for your evaluation.

Are the methods adequately described?

Yes

Thanks for your evaluation.

Are the results clearly presented?

Yes

Thanks for your evaluation.

Are the conclusions supported by the results?

Yes

Thanks for your evaluation.

3. Point-by-point response to Comments and Suggestions for Authors

Comments:

I read with great interest the novel retrospective research study showing CEUS predicting HIF-alpha and GLUT-1 in PDAC patients. This study complements the existing knowledge and has potential to prognosticate PDAC patients in the preoperative phase and such information may be crucial in treatment plans and recommendations for upfront resection versus neoadjuvant chemotherapy.

I have minor comments.

1, Table 5 why does HIF alpha have 36 - 36 groups but GLUT-1 has 33 and 39 groups?

2, You should discuss the drawback of CEUS being operator-dependent, needing special training and facilities/resources.

3, You should discuss the clinical implications of knowing that a PDAC patient has high HIF-alpha and GLUT-1. How does the care/treatment differ compared to not knowing this information? I do not see any discussion on the clinical application of this research.

4, As CEUS is not a good modality to image pancreas, is it possible that EUS can be done with SonoVue injection and pancreas imaged internally and if that would enhance the accuracy? or that would help identify areas to perform Needle biopsy? Anyone has done this? Some discussion is warranted.

5, Authors must mention some data about CA19-9 which is a tumour marker relevant in PDAC - PMID: 33437400

6, I do see authors mentioning some blood indices / ratios but there is minimal discussion about the utility of such indices in PDAC.

7, Authors have 100 over patient data about whipples / PDAC surgery and some operative details is necessary - how many whipples and how many distal panc, 30 and 90 day mortality, POPF, DGE, PPH and other relevant demographic details of the patients should be mentioned in tables

Thanks

Comments 1: Table 5 why does HIF alpha have 36 - 36 groups but GLUT-1 has 33 and 39 groups?

Response 1: Thank you for pointing this out. We apologize that we didn't elaborate it clearly. In the analysis of the correlation between CEUS quantitative perfusion characteristics and the expression of HIF-1α and GLUT1 (Table 5), only 72 cases obtained time-intensity curves (Table 5) among the 102 cases included in the above part (Table 1-4) of this study. The expression levels of HIF-1α and GLUT1 in these 72 cases were determined based on the median value of the 102 cases. The median IODsum in HIF-1α and GLUT1 staining of the 102 PDACs were 23955 and 20661, respectively. Therefore, IODsum value in HIF-1α  ≤ 23955 was defined HIF-1α low expression, IODsum value in HIF-1α>23955 was defined HIF-1α high expression. Among the 72 cases obtained standardized time-intensity curves, 36 cases had IODsum value in HIF-1α>23955, which were included in HIF -1α high group, and 36 cases had IODsum value in HIF-1α ≤ 23955, which were included in HIF -1α low group (36 - 36 groups). However, since the IODsum of HIF-1α and GLUT1 staining were different in each case, and the medians were also different, the expression levels (high or low expression) of HIF-1α and GLUT1 staining might be inconsistent one lesion. The median IODsum in GLUT1 staining of the 102 PDACs was 20661. Therefore, IODsum value in GLUT1 ≤ 20661 was defined GLUT1 low expression, IODsum value in GLUT1>20661 was defined GLUT1 high expression. Among the 72 cases obtained time-intensity curves, 33 cases had IODsum value in GLUT1≤ 20661, which were included in GLUT1 low group, and 39 cases had IODsum value in GLUT1>20661, which were included in GLUT1 high group (33 - 39 groups).  

The expressions of HIF-1α and GLUT1 in these 72 cases included in table 5 were as follows: 24 cases showed both HIF-1α-high and GLUT1-high expression, 21 cases showed both HIF-1α-low and GLUT1-low expression, 12 cases showed HIF-1α-high but GLUT1-low expression, and 15 cases showed HIF-1α-low but GLUT1-high expression (36 PDACs with HIF-1α-low expression, 36 PDACs with HIF-1α-high expression; 33 PDACs with GLUT1-low expression, 39 PDACs with GLUT1-high). We have supplemented these descriptions in the text on page 12, line 320-323.

Comments 2: You should discuss the drawback of CEUS being operator-dependent, needing special training and facilities/resources

Response 2: Thank you so much for pointing this out. We agree with this comment that the drawback of CEUS being operator-dependent, needing special training and facilities/resources should be discussed, and the limitation has been described in the discussion part of the text on page 17, line 508-513.

The contents are as follows:

Thirdly, CEUS is a highly operator-dependent imaging modality, needing specific training and equipment, and the imaging results may be suboptimal due to patient-related factors such as high BMI, bowel gas or ascites. In addition, pancreatic body or tail lesions are sometimes difficult to identify via transabdominal CEUS. The results of this study should be verified by CH-EUS and CH-EUS-FNA in the future.

Comments 3: You should discuss the clinical implications of knowing that a PDAC patient has high HIF-alpha and GLUT-1. How does the care/treatment differ compared to not knowing this information? I do not see any discussion on the clinical application of this research.

Response 3: Thank you for your valuable suggestions. We have supplemented some discussions on the clinical application of this research, which were mainly about the clinical implications of knowing that a PDAC patient has high HIF-1α and GLUT-1. Please find the details in the text on page 1-2, line 43-47, page 2, line 61-64, and page 15, line 371-388.

The contents are as follows:

On page 1-2, line 43-47: Hypoxia-targeted therapies provide new opportunities to overcome the resistance to chemotherapy and radiotherapy and enhance therapeutic efficacy, and a reliable imaging biomarker for hypoxia could help select appropriate patients and evaluate the therapeutic efficacy.

On page 2, line 61-64: HIF-1α play pivotal roles in hypoxia-induced therapy resistance of PDAC, and many hypoxia-targeted therapy works by inhibiting the expression of HIF-1α or HIF-1α-associated molecules such as glucose transporter-1 (GLUT1).

On page 15, line 371-388: The adaptive responses to hypoxic microenvironment stimulating a more aggressive and treatment-refractory phenotype of PDAC, and hypoxia may be the key point to improving current treatment strategies. [3,21]. Innovative therapies which targeting the tumor hypoxic hold great potential to overcome the chemoresistance and radioresistance and thus enhance the therapies efficacy [3]. It has been reported that high expression of HIF-1α in PDAC markedly reduced sensitivity to gemcitabine [22]. HIF-1α inhibitors can be used in high HIF-1α expression PDAC to improve therapeutic efficacy by inhibiting the overexpression of HIF-1α or promoting the degradation of HIF-1α [23,24]. Zhang et al. [25] have reported that inhibition of HIF-1α enhances the sensitivity to gemcitabine in PDAC via suppression of autophagic fux. Shukla et al. [26] have also suggested that targeting HIF-1α or HIF-1α-mediated metabolic pathways enhances the efficacy of gemcitabine. GLUT1 is a high-affinity glucose transporter that regulates glucose uptake with increased expression during hypoxia [27]. Glucomet-PDACs (PDAC with high glucose metabolism levels) have been reported to be resistant to chemotherapy, and pharmacological inhibition of GLUT1 enhances the chemotherapy response of glucomet-PDAC [28]. With a better understanding of the expression levels of HIF-1α and GLUT1, more effective and patient-specific therapies could be devised.

Comments 4: As CEUS is not a good modality to image pancreas, is it possible that EUS can be done with SonoVue injection and pancreas imaged internally and if that would enhance the accuracy? or that would help identify areas to perform Needle biopsy? Anyone has done this? Some discussion is warranted.

Response 4: Thank you for carefully reading our manuscript and pointing these out. In fact, CH-EUS (Contrast‐enhanced harmonic endoscopic ultrasound) makes it possible to combine EUS with contrast agent injection, achieving precise imaging of the interior of the pancreas. It can help to determine the ideal target area for EUS-FNA, avoiding necrotic areas and vessels within the tumor, which has been reported to be superior to standard EUS-FNA in pancreatic lesions. The results of this study can be verified by CH-EUS and CH-EUS-FNA owing to its superiority that allows detailed visualization of the intratumor blood flow of PDACs, precise sampling of target enhancement tissues, and point-to-point analysis of enhancement characteristics and tumor hypoxia status. These additions can be found in the text on page 16, line 443-451.

Comments 5: Authors must mention some data about CA19-9 which is a tumour marker relevant in PDAC - PMID: 33437400

Response 5: Thank you for your constructive comments. We have supplemented the data about CA19-9 in the study, and analyzed the correlation between CA19-9 and the hypoxic microenvironment of PDAC. However, no statistically significant difference was found between groups. In addition, we have also discussed the value of CA199 in PDAC. These additions can be found in Table 1, table 3, and in the text on page 15, and line 391-400.

The contents are as follows:

Carbohydrate antigen 19-9 (CA 19-9) is an important tumor marker for the diagnosis, management, and prognosis prediction of PDAC [31, PMID: 33437400]. Imamura et al. [32] have proven that elevated CA19-9 was one of the independent predictors for the early recurrence of PDAC after pancreatectomy. Kanda et al. [33] have reported that high CA19-9 PDAC demonstrated aerobic glycolysis enhancement. Wang et al. [34] have reported that the elevated serum CA19-9 is associated with a single nucleotide polymorphism (SNP) of the HIF-1α gene in PDAC, which is a DNA sequence variation and may be involved in oncogene expression. We tried to analyzed the relationship between CA19-9 level and the HIF-1α/GLUT1 expression of PDAC, but no significant differences were found between groups.

Comments 6: I do see authors mentioning some blood indices / ratios but there is minimal discussion about the utility of such indices in PDAC.

Response 6: Thank you for pointing this out. We agree with this comment that the discussion about the utility of these blood indices/ ratios in PDAC should be supplemented. In fact, these blood indices including systemic immune-inflammation index (SII), neutro-phil-lymphocyte ratio (NLR), and platelet-to-lymphocyte ratio (PLR) has been proven to be important prognostic indicators for PDAC, and also been reported could be used as biomarkers of response to neoadjuvant therapy in patients with PDAC. We have supplemented the discussion in the text on page 15, and line 400-407.

Comments 7: Authors have 100 over patient data about whipples / PDAC surgery and some operative details is necessary - how many whipples and how many distal panc, 30 and 90 day mortality, POPF, DGE, PPH and other relevant demographic details of the patients should be mentioned in tables.

Response 7: Thank you for your valuable suggestions to improve the quality of our manuscript. We have supplemented the demographic details of the patients such as surgery types, arterial or venous involvement, resection margin status, postoperative complications, 30 and 60 days mortality and other relevant demographic details of the patients in this study. Please find the details in the text on page 2-3, and line 84-99.

4. Response to Comments on the Quality of English Language

Point 1:  English language fine. No issues detected.

Response 1: Thanks for your Comments on the Quality of our English Language.

Sincerely,

Fang Nie

Reviewer 3 Report

Very interesting and important study. My comments:

1) The authors should explain on other potential causes of hypoenhancement, for example presence of tissue necrosis in the tumor

2) The authors should emphasize how their findings could explain some potential diagnostic implications of CH-US and CH-EUS, for example for tissue sampling thus directing the puncture in the hyper enhanced area of the lesion (on this regard cite the recent studies: PMID: 33481633 and PMID: 31031330)

3) The authors should comment eventual (if any) ongoing clinical studies on this topic

Author Response

For research article

Response to Reviewer 3 Comments

1. Summary

Thank you very much for taking the time to review this manuscript. Please find the detailed responses below and the corresponding revisions in the re-submitted files. All corrections are highlighted by using the 'track changes' option in Microsoft Word. In any case, we are open to consideration of any further comments and suggestions.

2. Questions for General Evaluation

Reviewer’s Evaluation

Response and Revisions

Does the introduction provide sufficient background and include all relevant references?

Yes

Thanks for your evaluation.

Are all the cited references relevant to the research?

Yes

Thanks for your evaluation.

Is the research design appropriate?

Yes

Thanks for your evaluation.

Are the methods adequately described?

Yes

Thanks for your evaluation.

Are the results clearly presented?

Yes

Thanks for your evaluation.

Are the conclusions supported by the results?

Yes

Thanks for your evaluation.

 3. Point-by-point response to Comments and Suggestions for Authors

Comments Very interesting and important study. My comments:

1) The authors should explain on other potential causes of hypoenhancement, for example presence of tissue necrosis in the tumor.

2) The authors should emphasize how their findings could explain some potential diagnostic implications of CH-US and CH-EUS, for example for tissue sampling thus directing the puncture in the hyper enhanced area of the lesion (on this regard cite the recent studies: PMID: 33481633 and PMID: 31031330).

3) The authors should comment eventual (if any) ongoing clinical studies on this topic.

Comments 1: The authors should explain on other potential causes of hypoenhancement, for example presence of tissue necrosis in the tumor.

Response 1: Thank you for your valuable suggestions. We have explained other potential causes of hypoenhancement such as intratumoral necrosis, severe fibrosis, or mucin, and supplemented it in the text on page 16, and line 440-441.

Comments 2: The authors should emphasize how their findings could explain some potential diagnostic implications of CH-US and CH-EUS, for example for tissue sampling thus directing the puncture in the hyper enhanced area of the lesion (on this regard cite the recent studies: PMID: 33481633 and PMID: 31031330).

Response 2: Thank you for point this out. We have emphasized how the findings of this study could explain some potential diagnostic implications of CH-US and CH-EUS, such as for tissue sampling, and cited the recent studies (PMID: 33481633 and PMID: 31031330) in our study. These contents can be found in the text on page 16, and line 441-456.

The contents are as follows:

EUS is an important diagnostic technique for pancreatic lesions and allows tissue sampling by means of fine-needle aspiration (FNA) and fine-needle biopsy (FNB) [41, PMID: 31031330]. CH-EUS (Contrast‐enhanced harmonic endoscopic ultrasound) makes it possible to combine EUS with contrast agent injection, achieving precise imaging of the interior of the pancreas. It can help to determine the ideal target area for EUS-FNA, avoiding necrotic areas and vessels within the tumor, which has been reported to be superior to standard EUS-FNA in pancreatic lesions [42, PMID: 33481633]. The results of this study can be verified by CH-EUS and CH-EUS-FNA owing to its superiority that allows detailed visualization of the intratumor blood flow of PDACs, precise sampling of target enhancement tissues, and point-to-point analysis of enhancement characteristics and tumor hypoxia status. In addition, our study can help to establish a method to verify the amount and exact locations of hypoxic of PDAC with CEUS or CH-EUS information during the tissue sampling. With targeted tissue sampling, hypoxic and potentially chemoresistant cancer tissue can be precisely obtain for further reveal hypoxia related alterations and identify key molecular of hypoxic tumors.

Comments 3: The authors should comment eventual (if any) ongoing clinical studies on this topic.

Response 3: Thank you for your valuable suggestions. Our study suggested that CEUS may become a noninvasive imaging biomarker for tumor hypoxic microenvironment of PDAC. Currently, there have been several ongoing clinical studies on the development of imaging biomarkers for tumor hypoxic. We have commented them in this study. In addition, we have also supplemented some comment about the clinical studies on this topic which have been published recently. Please find the details in the text on page 16-17, and line 457-482.

4. Response to Comments on the Quality of English Language

Point 1: English language fine. No issues detected.

Response 1: Thanks for your Comments on the Quality of our English Language.

Sincerely,

Fang Nie

Reviewer 4 Report

The authors showed that the characteristics of contrast-enhanced ultrasound correlate with the hypoxic microenvironment. Tumor size, lymph node metastasis, incomplete enhancement, and isoenhancement of PED were independent predictors of HIF-1α-high PDACs and GLUT1-high PDACs. The paper is generally well written. However, I want to raise several concerns.

1. Did all patients in the study undergo upfront surgery? How many patients undergo surgery after neoadjuvant chemotherapy? How many patients had resectable or borderline resectable disease? Please provide details about the preoperative status of patients.

2. Similarly, how many patients showed arterial or venous involvement preoperatively?

3. How many patients underwent R0, R1, and R2 resection? Are there any patients with incomplete resection? Is there any relationship between incomplete resection and the hypoxic microenvironment?

4. Line 23: Please correct the misspelling of “enhacement”.

5. Could you identify all cases of pancreatic body or tail cancer? Generally, pancreatic body or tail cancer is sometimes difficult to identify via transabdominal ultrasound.

6. How did the authors rule out metastasis? Which modality did the authors use? Please provide the details.

Minior editing is needed.

Author Response

For research article

Response to Reviewer 4 Comments

1. Summary

Thank you very much for taking the time to review this manuscript. Please find the detailed responses below and the corresponding revisions in the re-submitted files. All corrections are highlighted by using the 'track changes' option in Microsoft Word. In any case, we are open to consideration of any further comments and suggestions.

2. Questions for General Evaluation

Reviewer’s Evaluation

Response and Revisions

Does the introduction provide sufficient background and include all relevant references?

Yes

Thanks for your evaluation.

Are all the cited references relevant to the research?

Yes

Thanks for your evaluation.

Is the research design appropriate?

Can be improved

Thanks for your evaluation.

Are the methods adequately described?

Can be improved

Thanks for your evaluation.

Are the results clearly presented?

Can be improved

Thanks for your evaluation.

Are the conclusions supported by the results?

Yes

Thanks for your evaluation.

 3. Point-by-point response to Comments and Suggestions for Authors

Comments The authors showed that the characteristics of contrast-enhanced ultrasound correlate with the hypoxic microenvironment. Tumor size, lymph node metastasis, incomplete enhancement, and isoenhancement of PED were independent predictors of HIF-1α-high PDACs and GLUT1-high PDACs. The paper is generally well written. However, I want to raise several concerns.

1. Did all patients in the study undergo upfront surgery? How many patients undergo surgery after neoadjuvant chemotherapy? How many patients had resectable or borderline resectable disease? Please provide details about the preoperative status of patients.

2. Similarly, how many patients showed arterial or venous involvement preoperatively?

3. How many patients underwent R0, R1, and R2 resection? Are there any patients with incomplete resection? Is there any relationship between incomplete resection and the hypoxic microenvironment?

4. Line 23: Please correct the misspelling of “enhacement”.

5. Could you identify all cases of pancreatic body or tail cancer? Generally, pancreatic body or tail cancer is sometimes difficult to identify via transabdominal ultrasound.

6. How did the authors rule out metastasis? Which modality did the authors use? Please provide the details.

Comments 1: Did all patients in the study undergo upfront surgery? How many patients undergo surgery after neoadjuvant chemotherapy? How many patients had resectable or borderline resectable disease? Please provide details about the preoperative status of patients.

Response 1: Thank you for pointing this out. We apologize that we didn't provide details about the preoperative status of patients before. Among the 102 cases we included in this study, only 4 (3.9%) patients undergo surgery after neoadjuvant chemotherapy; 88 (86.3%) patients had resectable PDAC, and 14 (13.7%) patients had borderline resectable PDAC. We have supplemented these descriptions in the text on page 2, line 84-86.

Comments 2: Similarly, how many patients showed arterial or venous involvement preoperatively?

Response 2: Thank you for pointing this out. We apologize that we didn't elaborate them clearly before. Among the 102 cases we included in this study, 14 (13.7%) cases had vascular involvement, including solely splenic vein involvement in 6 cases, solely superior mesenteric vein involvement in 1 case, solely superior mesenteric artery involvement in 1 case, combined portal vein and splenic vein involvement in 2 cases, combined superior mesenteric vein and superior mesenteric artery involvement in 2 cases, and combined splenic vein and splenic artery involvement in 2 cases; We have supplemented these descriptions in the text on page 2, and line 87-91.

Comments 3: How many patients underwent R0, R1, and R2 resection? Are there any patients with incomplete resection? Is there any relationship between incomplete resection and the hypoxic microenvironment?

Response 3: Thank you again for your valuable suggestions to improve the quality of our manuscript. Among the 102 cases we included in this study, 92 (90.2%) patients underwent R0 resection, and 10 (9.8%) patients underwent R1 resection (Please find the details in the text on page 2, line 93-94). In addition, we have analyzed the relationship between the resection margin status (R0/R1) and the hypoxic microenvironment of PDAC, but no significant relationship was found between groups (Please find the details in table 1 and table 3). 

Comments 4: Line 23: Please correct the misspelling of “enhacement”.

Response 4: Thank you very much for pointing this out. We have corrected the misspelling of “enhacement” as “enhancement “on line 23.

Comments 5: Could you identify all cases of pancreatic body or tail cancer? Generally, pancreatic body or tail cancer is sometimes difficult to identify via transabdominal ultrasound.

Response 5: Thank you again for pointing this out. We agree that pancreatic body or tail cancer is sometimes difficult to identify via transabdominal ultrasound, particularly in patients with a high BMI or those with bowel gas or ascites, where the imaging results may be suboptimal. We have supplemented the limitation in the discussion part on page 17, line 508-513. However, in this study, all CEUS examinations were performed on the basis that the lesions (including lesions located at pancreatic body or tail) can be clearly displayed on the corresponding gray-scale ultrasound (US). We apologize that we didn't elaborate this point clearly before, and we have supplemented it in the text on page 3, and line 102-104. In addition, for the lesions (including lesions located at pancreatic body or tail) which were unqualified with CEUS images, we have excluded them from this study (page 2, line 83-84).

Comments 6: How did the authors rule out metastasis? Which modality did the authors use? Please provide the details

Response 6: Thank you for pointing this out. We rule out metastasis through radiological imaging modalities such as CT, PET-CT, MRI, and US, or biopsy pathology for the suspicious lesions. Among the 102 cases included in this study, no suspicious distant metastasis was found through imaging modalities before surgery. In addition, whether there was regional lymph node metastasis or not were determined through the pathological results, in which the specimens were obtained through intraoperative lymph node dissection.

4. Response to Comments on the Quality of English Language

Point 1: Minor editing of English language required.

Response 1: Thanks for your Comments on the Quality of our English Language. We have carefully checked and corrected the English spelling in the whole manuscript.

Sincerely,

Fang Nie

Round 2

Reviewer 3 Report

The revised version of the paper is OK. Thank you!